# Evaluation of Liver Quality after Circulatory Death versus Brain Death: A Comparative Preclinical Pig Model Study

**DOI:** 10.3390/ijms21239040

**Published:** 2020-11-27

**Authors:** Jérôme Danion, Raphael Thuillier, Géraldine Allain, Patrick Bruneval, Jacques Tomasi, Michel Pinsard, Thierry Hauet, Thomas Kerforne

**Affiliations:** 1Inserm U1082, F-86000 Poitiers, France; jerome.danion@chu-poitiers.fr (J.D.); raphael.thuillier@chu-poitiers.fr (R.T.); geraldine.allain@chu-poitiers.fr (G.A.); Thomas.KERFORNE@chu-poitiers.fr (T.K.); 2Faculté de Médecine et de Pharmacie, Université de Poitiers, F-86000 Poitiers, France; 3CHU de Poitiers, Service de Chirurgie Générale et Endocrinienne, F-86021 Poitiers, France; 4CHU Poitiers, Service de Biochimie, F-86021 Poitiers, France; 5CHU Poitiers, Service de Chirurgie Cardiothoracique et Vasculaire, F-86021 Poitiers, France; tomasi.jacques@gmail.com; 6Hôpital Européen Georges Pompidou, Service D’anatomie Pathologique, F-75015 Paris, France; patrick.bruneval@aphp.fr; 7Faculté de Médecine, Université Paris-Descartes, F-75006 Paris, France; 8CHU Poitiers, Service de Réanimation Chirurgie Cardio-Thoracique et Vasculaire, Coordination des P.M.O., F-86021 Poitiers, France; Michel.PINSARD@chu-poitiers.fr; 9Fédération Hospitalo-Universitaire SUPORT, F-86000 Poitiers, France; 10IBiSA Plateforme ‘Plate-Forme MOdélisation Préclinique—Innovation Chirurgicale et Technologique (MOPICT)’, Domaine Expérimental du Magneraud, F-17700 Surgères, France; 11Pr. Thierry HAUET, INSERM U1082, CHU de Poitiers, 2 rue de la Miletrie, CEDEX BP 577, 86021 Poitiers, France

**Keywords:** liver transplantation, Maastricht class III donor, brain death donor, ischemia reperfusion injury

## Abstract

The current organ shortage in hepatic transplantation leads to increased use of marginal livers. New organ sources are needed, and deceased after circulatory death (DCD) donors present an interesting possibility. However, many unknown remains on these donors and their pathophysiology regarding ischemia reperfusion injury (IRI). Our hypothesis was that DCD combined with abdominal normothermic regional recirculation (ANOR) is not inferior to deceased after brain death (DBD) donors. We performed a mechanistic comparison between livers from DBD and DCD donors in a highly reproducible pig model, closely mimicking donor conditions encountered in the clinic. DCD donors were conditioned by ANOR. We determined that from the start of storage, pro-lesion pathways such as oxidative stress and cell death were induced in both donor types, but to a higher extent in DBD organs. Furthermore, pro-survival pathways, such as resistance to hypoxia and regeneration showed activation levels closer to healthy livers in DCD-ANOR rather than in DBD organs. These data highlight critical differences between DBD and DCD-ANOR livers, with an apparent superiority of DCD in terms of quality. This confirms our hypothesis and further confirms previously demonstrated benefits of ANOR. This encourages the expended use of DCD organs, particularly with ANOR preconditioning.

## 1. Introduction

Transplantation is the most efficient therapy for end stage organ failure. While, organ surrogates exist for the kidney, it is not the case for other organs, particularly the liver and lung, for which transplantation is the only life-saving therapy available.

While advances in immunity, surgical techniques and patient care have improved liver transplantation (LT) success, there remains an important rate of complications [1]. Moreover, as with other organs, there is a critical liver shortage (in France in 2017, 2806 patients were waiting for a liver, for only 1374 transplantation performed), imposing a demographic change in the donor population (50% of donors over 50 years of age in Europe in 2009 [2]). Unlike the kidney, there is no consensus on the definition of marginal donor for the liver. A donor risk index has been developed, using prolonged cold ischemic time (CIT), age, steatosis, partial graft, positive serologies for Hepatitis B or C virus or human T cell leukemia/lymphoma virus type 1, female sex, high serum sodium (Na^+^ > 155), elevated transaminases, elevated bilirubin, prolonged down time and donation after circulatory death (DCD) [3,4]. Unfortunately, the rate of complications increases as more of these high-risk index organs are transplanted.

DCD donors are increasingly used in LT [5] in response to the shortage. These are impacted by a period of warm ischemia, and have higher rates of complication compared to organs from deceased after brain death (DBD) donors, especially primary non-function, arterial thrombosis and biliary ischemic stricture [6]. In addition, standard risk factors such as the duration of cold ischemia appear to have an even more significant effect on graft survival [7]. This leads to higher rates of graft failure and decreased patient survival [8], and encourages consideration of a score dedicated to this type of donor [9].

Ischemia Reperfusion Injury (IRI) is likely a key factor accounting for this increased complication rate [10], hence the need for improved donor and organ management compared to DBD [11]. Recently, kidney studies suggested the benefits of abdominal normothermic oxygenated recirculation (ANOR), for DCD donors, significantly improving outcome [12,13]. This procedure is also termed “in-vivo normothermic recirculation” [14], “extracorporeal support” [15], “extracorporeal membranous oxygenation” [16] or “normothermic recirculation” [12]. In LT, ANOR investigations showed measurable success [13,17,18,19,20]. However, these small-scale pilot studies cannot permit the necessary comparison of ANOR-DCD to DBD donors.

We recently adapted ANOR to the Large White pig [21,22], an animal very similar to humans [23]. Warm ischemia (WI) after cardiac arrest was limited to 30 min (national recommendations) and ANOR was run for 4 h before organ collection. We also developed a brain death model, simulating a slow building encephalic pressure until brain death, followed by 4 h of reanimation [24].

The objective of this study was thus to compare organs from both DCD and DBD, in similar subject and timeline conditions, hence, producing accurate data to gauge lesions. These were compared to Control, normal livers from pigs not subjected to any protocol other than anesthesia. Our hypothesis was that DCD combined with ANOR is not inferior to standard DBD.

## 2. Results

Livers were compared at collection (0 h) and during static preservation with HTK (Histidine Tryptophane Ketoglutarate solution) after 2, 4 and 6 h, to explore a timeline encountered in the clinic; 8 h, the maximal cold ischemia time acceptable in the clinic; and 24 h, a time point only available in experimental setting which we judged of interest on a cognitive point on view. The experimental design is summarized in Figure 1.

Energy content in the tissue was estimated by the ratio of ATP concentration over all nucleotides (Appendix A), demonstrating a rapid decrease (−60%) in the first two hours followed by a steady decrease (−20%) until the 6 h time point, after which the concentration was stable. There was no difference between the groups.

### 2.1. Histological Evaluation of Livers at Collection and during Preservation 

Blood stasis (Appendix AA) was absent in CTL group but observed in both DCD and DBD samples, although not reaching significance. Sinusoidal dilation (Appendix AB) was important (average score: 2) in all group, without differences between donor types. Ballooning degeneration (Appendix AC) was absent in CTL but observed in both DCD and DBD samples, with a trend towards higher levels in DBD. Hepatocyte Cell Death (Appendix AD) showed a similar pattern. Finally, vacuolation (Appendix AE) was very limited in the CTL throughout the preservation period. While some samples were positive in the DCD and DBD groups, the variability does not permit definite conclusions.

### 2.2. Analysis of Stress Markers between the Three Groups

#### 2.2.1. Inflammation 

We explored TNFα expression by RT-QPCR (Figure 2A). In the CTL group, expression tended to increase at 0 h, then further increase after 2 h (*p* < 0.05) and until 8h, then decreased at 24 h. A similar pattern of expression was determined in DCD livers, although not reaching significance. Interestingly, expression remained significantly lower in DBD livers.

#### 2.2.2. Oxidative Stress

RT-QPCR showed that expression of Nox2 mRNA (Figure 2B), coding for an element of the NADPH oxidase, remained normal in both CTL and DBD groups, while DCD livers displayed significantly decreased expression, at all time points. Moreover, expression of the superoxide dismutase (SOD) 2 transcript (Figure 2C), coding for the mitochondrial Mn^2+^ SOD, was significantly increased in DBD, and further increased in DCD organs. Finally, we recorded a significantly increased expression of Heme Oxygenase-1 (HO−1, Figure 2D) in the CTL group after 2 and 8 h of cold ischemia. This response was not found in either DBD or DCD livers.

Presence of reactive oxygen species (ROS) was assayed on liver samples collected at time of collection (Figure 2, Bottom). We detected a minimal amount of signal in CTL (less than 3%) and a moderate induction of ROS production in DBD livers (7%). The highest levels of ROS production were detected in DCD samples, reaching 14% (*p* < 0.05 to CTL).

Western Blot analysis of oxidative stress proteins at time of collection showed significantly increased levels of SOD 1 (cytosolic) and 2 (mitochondrial) in DCD livers compared to CTL (Figure 3A,B). There was an increased level of catalase in both DBD and DCD groups compared to CTL (Figure 3C). Moreover, Heme oxygenase 1 (HO1) was significantly elevated in DCD compared to both CTL and DBD organs (Figure 3D). Finally, marker of lipid peroxidation 4-Hydroxynonenal (4-HNE) and protein nitrosylation nitrotyrosine (*N*–Tyr) were significantly elevated in DCD compared to CTL; while there was a trend towards increased levels in DBD organs (Figure 3E,F).

#### 2.2.3. Apoptosis

To determine the impact of pre-collection events on programmed cell death, we explored the expression of effectors of apoptosis. BclXL and MCL1 are members of the BH (Bcl-2 homology) 1–4 domain anti-apoptotic proteins, restricting the action of BH1–3 proteins such as Bax. While, BclXL was unchanged in the CTL liver, there was a trend towards an early increase in DCD organs, reaching significance at 24 h. On the other hand, DBD livers consistently displayed significantly decreased levels of expression (Figure 4A). MCL1 was unchanged in the CTL organs, and significantly increased in DCD livers at all time points. In DBD organs, the expression of MCL1 significantly increased at 0 h and appeared to remain increased, however it was consistently lower than in DCD organs (Figure 4B). Finally, the expression of clusterin, another apoptosis modulator, significantly decreased in DBD livers, while it remained normal in both CTL and DCD organs (Figure 4C).

We measured the level of observable apoptosis at collection by TUNEL (Figure 4, right). There was only a few positive cells in the CTL group, while the DCD livers showed a significantly increased signal. DCD livers showed a trend towards increased apoptosis. Interestingly, positive cells in DCD livers were re-grouped in limited areas, while in DBD the cells were dispersed in the tissue.

We measured the ratio of Bax/Blc2 protein levels (Appendix A). There was no difference between the CTL and DBD groups, but there was a significant reduction in DCD organs, suggesting lower mitochondrial apoptosis pathway activation.

### 2.3. Analysis of Survival Pathways

#### 2.3.1. Hypoxia

Western blot showed that expression of Hypoxia Inducible Factor 1 α (HIF1α) was significantly increased in the DCD livers compared to both CTL and DBD (Figure 5A). RT-QPCR showed that in both, the CTL and the DCD groups, HIF1α mRNA remained unchanged, while in DBD the expression appeared decreased at 0, 2 and 8 h (Figure 5B). Regarding HIFα downstream targets; for both GLUT1 and NOTCH4 (Figure 5C,D), levels in the DBD livers were significantly decreased compared to the two other groups. No differences were recorded between the time points within groups.

#### 2.3.2. Liver Regeneration

The liver is unique in its ability to regenerate, a pathway involving the epidermal growth factor (EGF) receptor, to which can bind TGFα (transforming growth factor α) and HBEGF (heparin-binding EGF-like growth factor). Herein, we determined that both these factors were expressed during CS at 2 and 8 h, regardless of the group (Figure 6A,B). TGFβ, another mediator of liver regeneration, also showed increased expression levels in the DCD group at 2 and 8 h of CS, with CTL livers showing a similar trend. However, TGFβ expression was significantly lower in DBD organs (Figure 5C). Expression of hepatocyte growth factor (HGF) increased in DCD livers, and severely decreased in DBD (Figure 6D). Expression of WNT4 was decreased in the DBD group compared to CTL or DCD (Figure 6E). Finally, erythropoietin (EPO) expression was unchanged in the CTL group, but was enhanced in the DBD group and further increased in the DCD livers (Figure 6F). Western blot analysis of EPO showed that there was not, at H0, measurable differences between the groups (Appendix A).

## 3. Discussion

We previously demonstrated that ANOR could counteract DCD-induced IR lesions [22]. Herein, we investigated the hypothesis that DCD organs, when ANOR was used, were better than marginal organs and actually non inferior to standard DBD [24]. We tested this in a fully controlled preclinical model.

We first showed that during preservation there was an unsurprising decrease in ATP concentration, consistent with the effects of hypoxia and hypothermia on energy metabolism [25]. As this was no different between groups, we endeavored to investigate the know IR mechanisms.

Histological evaluation did not reveal differences between groups. This appears to contradict previous studies, which demonstrated vacuolation as a good marker of quality in DCD livers [26]. However, in our case, DCD donors were managed with ANOR, which confirms that this procedures enhances organ quality over standard DCD and should be deployed for these donors, as is the policy in France.

Typically associated with deleterious immune response, TNFα is also a signal for finely regulated apoptosis, removing unhealthy cells without excessive production of danger signals. In this light, our results suggest that CTL and DCD livers are able to mount a controlled response to the stresses of CS, while DBD organs do not. This pattern will be further evidenced in the remainder of the data. Other inflammation markers, Toll Like Receptor 2, *p* selectin, EGR1 or IL6, did not show changes (data not shown). This may appear counter-intuitive considering the consequences of brain death on inflammation [27], however at the organ level we may observe the consequence of systemic inflammation.

Oxidative stress is found in many physiopathological processes, including hepatic IRI [28]. Herein, DCD protocol induced a significant increase in ROS production. Markers of lipid peroxidation and protein nitrosylation were increased, as confirmed using CellRox. However, this was concomitant with increased anti-oxidant proteins expression: SOD 1 and 2, catalase and HO-1. Hence it is likely that the warm ischemia increased reactive oxygen species production, but that these organs are able to mount a response, a protection likely linked to the use of ANOR [22]. This observation is strengthened by the decreased levels of Nox2 mRNA in DCD organs. Nevertheless, the DCD protocol appears to be more conducive to oxidative stress, a parameter, which should be taken into account when considering employing these organs in transplantation, for instance with the use of anti-oxidative stress strategies [29].

DBD organs showed a lesser activation of markers of stress and anti-oxidative proteins, indicating that this injury may not be as much a concern in brain death donors. Interestingly, the expression of HO-1 increased during storage in CTL organs, while it remained at normal levels in DBD and DCD livers. This suggests that the liver, if healthy, is able to respond to the stresses of hypothermic ischemia by producing protective proteins against oxidative stress.

We then investigated apoptosis, another key pathway of liver IRI. DBD consistently demonstrated low expression levels for MCL1 and BclXL, both members of the BH1-4 protein family, negatively regulating apoptosis induction. The expression of Clusterin, a pro-survival protein linked to apoptosis modulation [30], followed a similar pattern at collection and was not altered by preservation. Therefore, DBD livers demonstrated distinct pro-apoptotic programming, compared to DCD and CTL. TUNEL analysis showed that, in accordance with transcriptomic data, the number of apoptotic cells in DBD livers significantly increased, compared to the other groups. While, there were positive cells in DCD organs, the staining was localized to specific areas, suggesting a higher level of control on programmed cell death, in line with the transcriptomic data. Decreased apoptosis in DCD livers may be due to lower mitochondrial pathway apoptosis, as shown by our Bax/Bcl2 ratio results, suggesting better mitochondria preservation and optimal functionality. Finally, lower detection of apoptosis in DCD liver may be because sampling took place 4 h after warm ischemic injury, and that during ANOR the organ was able to initiate repair mechanisms. This is concordant with the observed dynamic nature of survival mechanisms observed in another ANOR study, albeit focusing on the kidney [22]. Further investigation of the behavior of liver tissue during ANOR is required to investigate this aspect.

Investigating hypoxia, DCD livers expressed high levels of HIF1α, denoting a response to low oxygen levels. We again found similitudes between CTL and DCD liver, with increased to normal expression of markers such as HIF1α and its targets GLUT1 and NOTCH 4, while DBD livers systematically displayed lower expressions. This remained unchanged during CS, suggesting the involvement of events prior to organ collection. Hence, brain death appears deleterious to the organ, in regards to its ability to withstand hypoxic stress, while livers collected in DCD post-ANOR retain the ability to express hypoxia-resistance transcripts. This, combined with the apoptosis data, strongly suggests that ANOR preconditions the organ to better manage reperfusion injury, with an enhanced ability to manage cell death, as well as to restart energy metabolism and nutrient influx through angiogenesis.

Regeneration markers investigations demonstrated two patterns: while some markers were expressed in all livers (TGFα, HBEGF), DBD livers showed significantly lower expression levels for others (TGFβ, HGF, WNT4). Moreover, the differences observed for the latter appears independent on cold ischemia time, while the former were induced by ischemia. Therefore, it appears that events pre-collection affect cells so that they cannot express the required growth factors and mediators of cell division when faced with CS stresses. Here, it also appears that brain death impedes the ability of the organ to respond to the stresses of preservation.

EPO is generally not found in the liver, except during embryonic development [31]. However, we detected a production of EPO mRNA in both DBD and DCD organs. EPO proteins could be detected in all groups at collection, without differences between the groups, suggesting a post-transcriptional regulation. Interestingly, this EPO expression could be linked with hypoxia or associated with the recapitulation of developmental pathways.

Histological analysis did not reflect the differences between the groups evidenced through the other techniques. Indeed, the level of injury was similar between groups, and limited in intensity. This may be due to the fact that scoring was performed during preservation, a time point which other teams have demonstrated is not adequate for histological discrimination [32].

Interestingly, our transcriptomics data highlight that mRNA production is possible during CS (TNFα, HO1, HBEGF or TGFα). In alignment with other studies [33], there is indeed an active metabolism in place. This implies that: 1-CS is not a stasis or a slowed decrepitude, but an active stage in the life of the organ; 2-if transcriptome programming alteration takes place, it can be altered and directed through external means [25], especially with perfusion machines that can deliver the treatment inside the organ in a continuous manner.

Our study included a follow up of the organs until 24 h of preservation. While, having little relevance to the clinical setting, it showed that on the one hand, transcripts altered by protocols pre-collection remained unaffected by cold ischemia time (such as EPO, MCL1, SOD2). While, on the other hand, transcripts modulated by cold ischemia, such as TNFα, HO1, HIF1α, GLUT1 and TGFα, were back to normal levels at 24 h. This highlights that while expression of CS resistance mediators is possible, it is limited in time and extended cold ischemia time depletes energy and restricts RNA expression. Therefore, targeting this depletion could improve storage length and organ quality. Moreover, this dynamic process could be used to determine organ quality as differential expression was detected according to organ quality (CTL, the highest quality possible, versus DBD and DCD organs). 

In France, livers from DCD donors are increasingly used for transplantation (ABM2018), and this trend is growing in Europe. While, organ quality appears equivalent to standard DBD, it is yet too soon to establish definite conclusions. It is likely that these good outcomes are strongly associated with the required use of ANOR for DCD donation in France, which tends to agree with our results. Thus, it appears critical to explore the possibilities offered by ANOR optimization, particularly since DCD donor criteria expansion is discussed. 

Furthermore, optimization efforts need to extend further than ANOR. Indeed, liver perfusion machines are currently tested for marginal donation (HOPExt in France, ClinicalTrials.gov NCT03929523), following positive results from a Netherland study [34]. In fact, a randomized control trial is ongoing for DCD donors (DHOPE-DCD ClinicalTrials.gov NCT02584283 [35]). Such strategies are promising, and efforts are being made towards their optimization as well [36], and open key perspectives in organ quality determination, monitoring and even repair [37,38].

The dynamic nature of processes, within the preserved organ, is indeed an issue in transplantation, which led teams to develop more adapted preservation using normothermic circulation [39,40]. However, as normothermia presents a higher risk level, coupled with increases in complexity, it may be more appropriate to find an intermediate temperature. It should also have a management regiment targeted towards well-characterized mechanisms, such as highlighted in our study as well as others [41], in order to combine the best of both approaches. Nevertheless, these recent studies present highly interesting perspectives in terms of organ evaluation, with a longer preservation time, which permits an in depth evaluation of the organ, highlighting preservation as a unique window of opportunity for specific treatment of the graft, with therapies directed towards key injury pathway [36]. These studies rekindle the concept of organ hub across all transplanted organs, a priority in the field.

Our study presents several limitations. (1) DCD is coupled with ANOR and we did not compare it with DCD without ANOR. However, current clinical [13,17,18] and pre-clinical [22] studies have clearly demonstrated the advantages of performing ANOR for DCD, such as it has become required practice in France (Agence de la Biomédecine Guidelines). Hence, DCD without ANOR does not seem to be relevant and we chose not to include it so as to follow the mandatory ethical regulations. (2) We did not transplant the preserved liver, in order to rigorously gauge the model regarding the precise timing of the procedure, again following bioethics regulations. Indeed, if the present preliminary study, extensively investigating the mechanisms underlying the three donor types, did not confirm our hypothesis of non-inferiority of DCD over standard DBD, there was not need to use more animal. While this is limiting, the originality of our study is that for the first time all three donor types are performed in parallel, with identical baseline conditions, allowing for accurate comparison rather than inference with data from the literature. In addition, this approach allowed us to optimize and present the most accurate preclinical models for these donor types. (3) True DCD donation involves withdrawal of treatment, an agonal phase characterized by haemodynamic, catecholamine release and ischemic injury before asystole [42,43,44]. Reproducing in the pig is a very complex procedure, and the present study is anterior while we are developing this aspect [45]. One could also consider a limitation the fact that on several western blots (HO1, SOD1, SOD2, and HIF1) the protein signal, while different, does appear close between DCD and DBD and that the recorded difference is due to less total protein as showed by the use of StainFree technology. While confusing, this highlight the value of StainFree over classical housekeeping protein normalization: StainFree evaluates protein loading over the whole gel rather than basing it on a single protein that often is affected by the protocol.

Altogether, our data highlights critical differences between DBD and ANOR-DCD livers. Indeed, regarding lesions pathways such as oxidative stress and cell death, and stress resistance pathways such as hypoxia and regeneration, ANOR-DCD organs were superior to DBD livers. Hence, beyond non-inferiority, we demonstrate superiority of ANOR-DCD organs over DBD. Our study also encourages an in depth investigation of ANOR timing focused on the liver, as fine tuning ANOR duration could further enhance the quality of the organ, as demonstrated for the kidney and its role in conditioning and repair [22]. Moreover, we highlight several pathways which could be used to develop new, and necessary, approaches to improve transplantation outcome: Risk stratification through quality evaluation, and/or targeted therapeutics applied per-transport [46], possibly combined with machine perfusion [47]. As the organ shortage is growing, expanding the use of ANOR-DCD organs appears a source of organs, possibly combined with other conditioning strategies [48]. The scientific community has big challenges ahead of it and a great deal of research is still needed to optimize liver preservation. In the future, we may progress toward organ-tailored preservation for livers and particularly high-risk ones opening the way to a new era and to conduct a predictive approach and to validate predictive tools.

## 4. Materials and Methods

Detailed methods can be found in the Appendix A: Experimental model, Anesthesia, ANOR model, Brain death model, Organ collection, Experimental design and sample collection, Anatomopathological analysis, Real-Time Quantitative PCR, Western blot analysis, TUNEL Staining, CellRox Staining.

Animal experiments were conducted at the MOPICT platform (Surgeres, France) in accordance with the Animal research guidelines, the French Government, and the instutitional Committee on the Ethics of Animal Experiments (France) (Committee accreditation number: C2EA-84, Protocol approval numbers: CE2012-11 (DCD model) and CE2012-14 (DBD model).

### 4.1. DCD/ANOR Model 

Circulatory arrest was induced by the injection of 2 g potassium chloride through the central venous catheter then the mechanical ventilation was stopped [21,22]. After 30 min of WI, ANOR was started for 4 h before organ collection.

### 4.2. Brain Death Model 

Brain death was reached by progressive insufflation of a balloon in the extradural space until a flat electroencephalogram was obtained [24]. Reanimation was maintained for 4 h before organ collection.

### 4.3. Experimental Design and Sample Collection 

Livers were collected after each of the three protocols and kept at 4 °C during a period of 24 h (Appendix A: Experimental set-up).

### 4.4. Anatomopathological Analysis

HES and PAS stained slides were evaluated by a blinded pathologist. Semi-quantitative evaluation of lesions was performed as detailed in Appendix A.

### 4.5. Real-Time Quantitative PCR

Primer sequences are detailed in Appendix A. mRNA expression levels in the samples relative to expression in healthy control kidney were determined with the Pfaffl method [49] (expressed as Relative Fold Change to Control livers).

### 4.6. Western Blot Analysis

Western Blotting was carried out according to standard protocols.

### 4.7. TUNEL Staining

TdT-mediated dUTP Nick-End Labeling assay was performed on frozen samples using the DeadEnd™ Fluorometric TUNEL System according to the manufacturer’s recommendations. 

### 4.8. CellRox Staining

Production of reactive oxygen species was measured on frozen samples using CellROX^®^ Green Reagent.

### 4.9. Statistical Analysis

Five animals were used in each group (*n* = 5). Results are expressed as mean ± SEM (standard error mean). Multiple comparison tests were performed using the Kruskal Wallis test with post hoc Dunn’s Test, on the NCSS software (NCSS LLC, Kaysville, UT, USA). Statistical significance was set at *p* < 0.05.

## Figures and Tables

**Figure 1 ijms-21-09040-f001:**
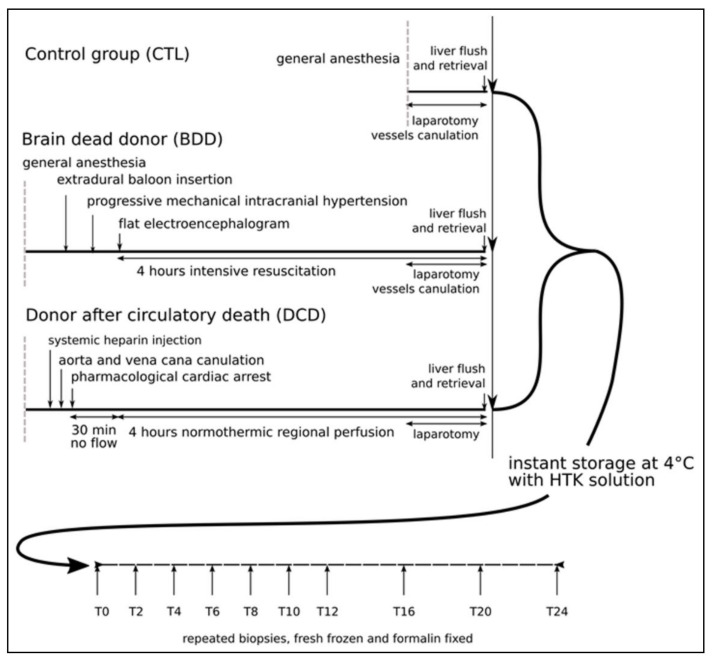
**Experimental set-up**: The control group (CTL) consisted in a laparotomy followed by liver removal. The brain death donor group (DBD) demonstrated a flat electroencephalogram as a result of cranial hypertension and was followed by resuscitation for 4 h. The donation after circulatory death group (DCD) began with the insertion of cannulas into the aorta and vena cava, then was euthanized during anesthesia by potassium overdose and followed by a 4-h regional normothermic circulation. The three types of graft were subsequently flushed and conditioned according to the same protocol at 4 °C with repeated biopsies allowing for further analysis. Five animals were used in each group (*n* = 5).

**Figure 2 ijms-21-09040-f002:**
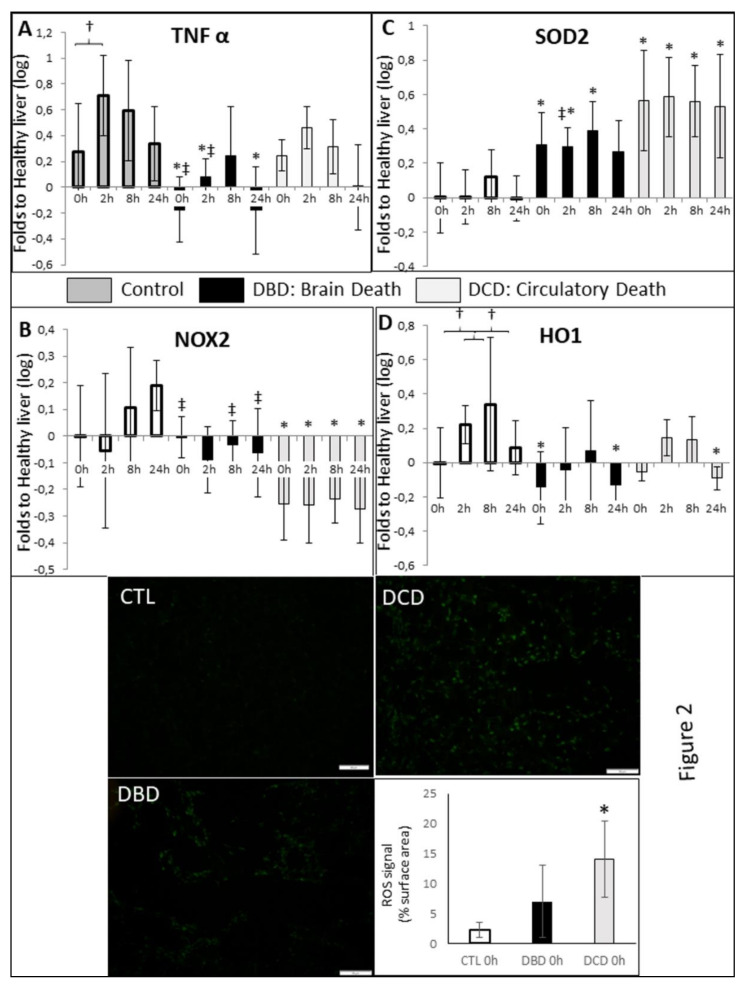
**Evaluation of inflammation and oxidative stress markers:***Top: RTqPCR*. Frozen samples were collected at the indicated times and processes for RTqPCR analysis. mRNA expression levels of (**A**): TNF α, (**B**): NOX2, (**C**): SOD2 and (**D**): HO1 in the samples relative to expression in healthy control liver were determined with the Pfaffl method (expressed as Relative Fold Change), using ribosomal L19, α-Actin, SDHA, CYA62 and RPLPO genes as internal controls. Healthy liver were chosen as normalization in order to gauge alterations in the Control group. *Bottom*: Histofluorescence. Frozen samples were collected after organ rinsing and processes for CellROX^®^ Green Reagent reaction. Representative images are shown from each group and quantification is displayed. Signal quantification was performed on 10–15 high powered fields (200×) using ImageJ. CTL group: white bars; DBD group: black bars; DCD group: gray bars. Statistics: shown are mean ± SEM, multiple comparison tests were performed using ANOVA Kruskal Wallis + Dunns Test, †: *p* < 0.05 within group; *: *p* < 0.05 to CTL at the same time; *p* < 0.05 ‡: to DCD, at the same time. *n* = 5.

**Figure 3 ijms-21-09040-f003:**
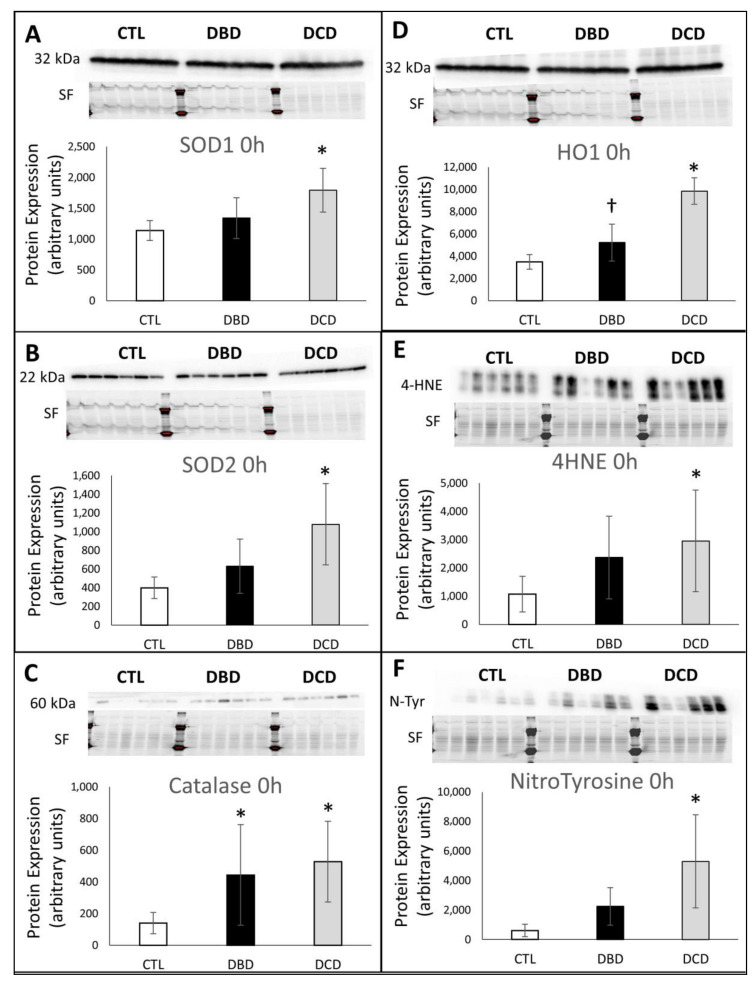
**Western Blot evaluation of inflammation and oxidative stress markers**: (**A**): SOD1, (**B**): SOD2, (**C**): Catalase, (**D**): HO-1, (**E**): 4-HNE, (**F**): NitroTyrosine. Frozen samples were collected at the indicated times and processes for western blot analysis. 30 µg protein were loaded for each lane. Each lane corresponds to the extract from 1 sample, at the indicated time. For each target, the signal is shown on top, with normalizing stain free blot (SF) shown underneath; quantification is shown at the bottom. CTL group: white bars; DBD group: black bars; DCD group: gray bars. Statistics: shown are mean ± SEM, multiple comparison tests were performed using ANOVA Kruskal Wallis + Dunns Test, *: *p* < 0.05 to CTL at the same time; †: *p* < 0.05 to DCD, at the same time. *n* = 5.

**Figure 4 ijms-21-09040-f004:**
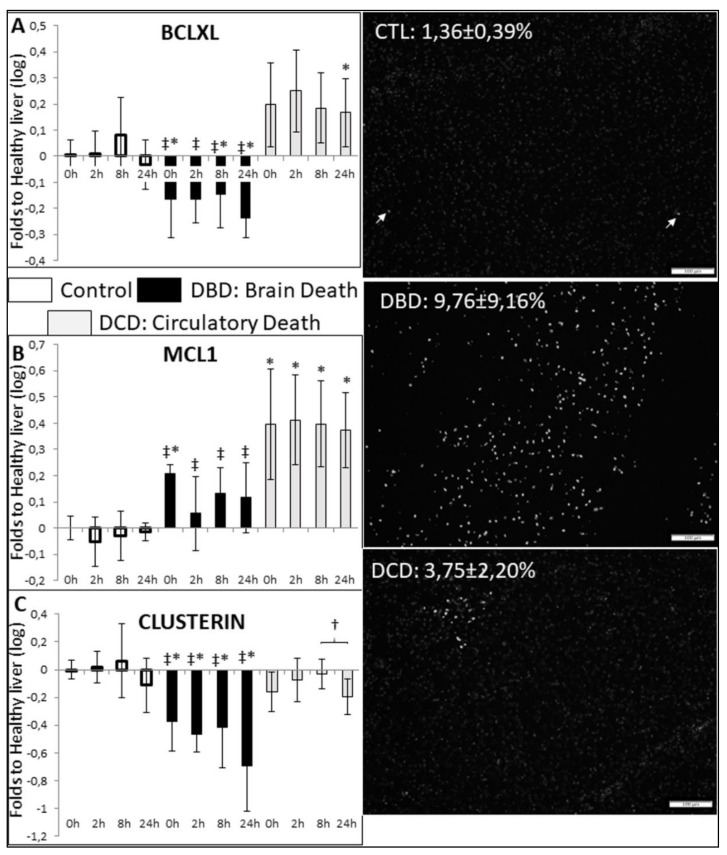
**RTqPCR and Histofluorescent evaluation of apoptosis**: Left: mRNA expression levels of (**A**): Bcl-XL, (**B**): MCL-1 and (**C**): Clusterin. Frozen samples were collected at the indicated times and processes for RTqPCR analysis. mRNA expression levels in the samples relative to expression in healthy control liver were determined with the Pfaffl method (expressed as Relative Fold Change), using ribosomal L19, β-Actin, SDHA, CYA62 and RPLPO genes as internal controls. Healthy livers were chosen as normalization in order to gauge alterations in the Control group. Right: Frozen samples were collected after organ rinsing and processes for TUNEL reaction. Representative images are shown from each group and quantification is displayed. Signal quantification was performed on 10–15 high-powered fields (200×) using ImageJ, cells were considered positive when the nucleus was clearly white (arrows). CTL group: white bars; DBD group: black bars; DCD group: gray bars. Statistics: shown are mean ± SEM, multiple comparison tests were performed using ANOVA Kruskal Wallis + Dunns Test, †: *p* < 0.05 within group; *: *p* < 0.05 to CTL at the same time; *p* < 0.05 ‡: to DCD, at the same time. *n* = 5.

**Figure 5 ijms-21-09040-f005:**
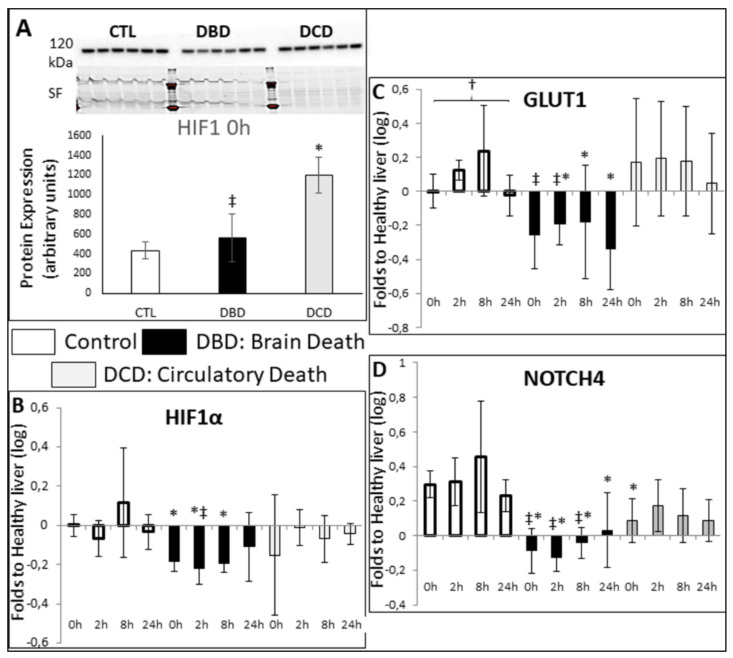
**Western blot and RTqPCR evaluation of hypoxia markers**: Frozen samples were collected at the indicated times and processes for western blot (**A**) and RTqPCR analysis (**B**–**D**). (**A**): Western Blot analysis of HIF-1α protein expression. Frozen samples were collected at the indicated times and processes for western blot analysis. 30 µg protein were loaded for each lane. Each lane corresponds to the extract from 1 sample, at the indicated time. The signal is shown on top, with normalizing stain free blot (SF) shown underneath; quantification is shown at the bottom. (**B**–**D**): mRNA expression levels of (**B**): HIF-1α, (**C**): GLUT1 and (**D**): Notch4. mRNA expression levels in the samples relative to expression in healthy control liver were determined with the Pfaffl method (expressed as Relative Fold Change), using ribosomal L19, β-Actin, SDHA, CYA62 and RPLPO genes as internal controls. Healthy livers were chosen as normalization in order to gauge alterations in the Control group. CTL group: white bars; DBD group: black bars; DCD group: gray bars. Statistics: shown are mean ± SEM, multiple comparison tests were performed using ANOVA Kruskal Wallis + Dunns Test, †: *p* < 0.05 within group; *: *p* < 0.05 to CTL at the same time; *p* < 0.05 ‡: to DCD, at the same time. *n* = 5.

**Figure 6 ijms-21-09040-f006:**
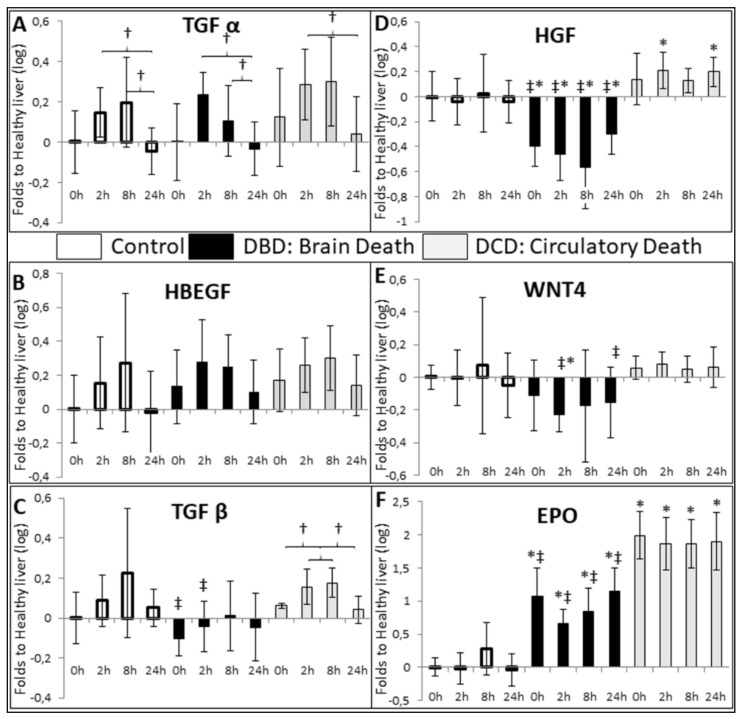
**RTqPCR Evaluation of regeneration markers**: Frozen samples were collected at the indicated times and processes for RTqPCR analysis. (**A**): TGFα, (**B**): HBEGF, (**C**): TGFβ, (**D**): HGF, (**E**): WNT4 and (**F**): EPO. mRNA expression levels in the samples relative to expression in healthy control liver were determined with the Pfaffl method (expressed as Relative Fold Change), using ribosomal L19, β-Actin, SDHA, CYA62 and RPLPO genes as internal controls. Healthy livers were chosen as normalization in order to gauge alterations in the Control group. CTL group: white bars; DBD group: black bars; DCD group: gray bars. Statistics: shown are mean ± SEM, multiple comparison tests were performed using ANOVA Kruskal Wallis + Dunns Test, †: *p* < 0.05 within group; *: *p* < 0.05 to CTL at the same time; *p* < 0.05 ‡: to DCD, at the same time. *n* = 5.

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
