# Peer review of "Evaluation of Liver Quality after Circulatory Death versus Brain Death: A Comparative Preclinical Pig Model Study"

_ijms, 2020, doi:10.3390/ijms21239040_

Round 1

Reviewer 1 Report

Danion et al. evaluated the liver quality after circulatory death versus brain death and found that DCD preconditioned with ANOR is not inferior to DBD by comparing both pro-lesion pathways and pro-survival pathways in a preclinical pig model. The conclusion is exciting considering the shortage of liver donors and may bring a profound impact in the field of transplantation. I would recommend the publication of this manuscript on IJMS if the following questions were addressed.

  1. I appreciate that all the three donor types were performed in parallel. With only one pig representing each donor type, however, it may not necessarily guarantee an accurate comparison due to potential variations between individuals.
  2. I wonder, in the opinion of the authors, when ANOR-DCD organs can be accepted in human patients. If not now, what further experiments should be performed. Or some more measures can be developed to improve it?

Author Response

Danion et al. evaluated the liver quality after circulatory death versus brain death and found that DCD preconditioned with ANOR is not inferior to DBD by comparing both pro-lesion pathways and pro-survival pathways in a preclinical pig model. The conclusion is exciting considering the shortage of liver donors and may bring a profound impact in the field of transplantation. I would recommend the publication of this manuscript on IJMS if the following questions were addressed.

  1. I appreciate that all the three donor types were performed in parallel. With only one pig representing each donor type, however, it may not necessarily guarantee an accurate comparison due to potential variations between individuals.

We thank the reviewer for his comment. However, there seem to have been a confusion, as 5 animals were used in each group, not one, as mentioned in the methods. We added this to the Figure 1 legend in order to dispel any possible confusion from the reader.

  1. I wonder, in the opinion of the authors, when ANOR-DCD organs can be accepted in human patients. If not now, what further experiments should be performed. Or some more measures can be developed to improve it?

Thank you for your comment, we indeed improved the discussion to touch on these points and related issues:

“In France, livers from DCD donors are increasingly used for transplantation (ABM2018), and this trend is growing in Europe. While organ quality appears equivalent to standard DBD, it is yet too soon to establish definite conclusions. It is likely that these good outcomes are strongly associated with the required use of ANOR for DCD donation in France, which tends to agree with our results. Thus, it appears critical to explore the possibilities offered by ANOR optimization, particularly since DCD donor criteria expansion is discussed.

Furthermore, optimization efforts needs to extend further than ANOR. Indeed, liver perfusion machines are currently tested for marginal donation (HOPExt in France, ClinicalTrials.gov NCT03929523), following positive results from a Netherland study [34]. In fact, a randomized control trial is ongoing for DCD donors (DHOPE-DCD ClinicalTrials.gov NCT02584283 [35]). Such strategies are promising, and efforts are being made towards their optimization as well [36], and open key perspectives in organ quality determination, monitoring and even repair [37,38].”

Reviewer 2 Report

This is an interesting study comparing the donor liver of DBD and ANOR-DCD using a porcine model. However, there are some problems and questions that need to be addressed.

1) The purpose of study is to show that ANOR-DCD is not inferior to deceased after DBD donors, and the authors concluded that ANOR-DCD was superior to DBD as a donor liver in the study. The conclusion is not completely acceptable for at least the following three reasons:

  1. The higher ROS signal in ANOR-DCD than in DBD may indicate a high degree of tissue damage in Fig 2.
  2. In the TUNEL assay in Fig. 4, ANOR-DCD has fewer positive cells than DBD, but TUNEL show an early reaction of apoptosis and it is possible the number of positive cells in ANOR-DCD had already begun to decrease at 4 hour.
  3. It seems that there is not much difference in band concentration between ANOR-DCD and DBD in HO1, SOD1, SOD2, and HIF1 in Fig. 3 and Fig. 5. Lower SF protein levels in ANOR-DCD than DBD may emphasize the increased protein expression.

It would be better to elaborate on the interpretation of these results to support the conclusion.

2) In Supplementary Figure 2, there is considerable variability in histological evaluation using HE. Though it is mentioned in limitation of discussion part, the fact that the effects of reperfusion injury have not been evaluated is also considered to be a reason why this conclusion is difficult to accept. However, it is impressive that BCLXL, GLUT1 and NOTCH4 at the mRNA level during cold preservation show converse results between DBD and ANOR-DCD. Please explain whether the difference can be associated with the superiority of ANOR-DCD.

3) Does the authors think that ANOR after DCD up-regulates stress proteins such as HO1 and HIF1 while DBD is less likely to upregulate them. If so, it would be better to examine the time-dependent change of the stress protein before donation of 4 hours. Is 4 hours normothermic regional perfusion optimal in the examination so far?

4)Does "H0" in Fig3 mean "0h"?

Author Response

This is an interesting study comparing the donor liver of DBD and ANOR-DCD using a porcine model. However, there are some problems and questions that need to be addressed.

 1) The purpose of study is to show that ANOR-DCD is not inferior to deceased after DBD donors, and the authors concluded that ANOR-DCD was superior to DBD as a donor liver in the study. The conclusion is not completely acceptable for at least the following three reasons:

  1. The higher ROS signal in ANOR-DCD than in DBD may indicate a high degree of tissue damage in Fig 2.

Thank you for this comment. We indeed completed our discussion of these results as follows: “Nevertheless, the DCD protocol appears to be more conducive to oxidative stress, a parameter which should be taken into account when considering employing these organs in transplantation, for instance with the use of anti-oxidative stress strategies “

  1. In the TUNEL assay in Fig. 4, ANOR-DCD has fewer positive cells than DBD, but TUNEL show an early reaction of apoptosis and it is possible the number of positive cells in ANOR-DCD had already begun to decrease at 4 hour.

We thank the reviewer for this insight. Indeed, decrease of apoptosis could have taken place during ANOR. We discussed this aspect as follows: “Finally, lower detection of apoptosis in DCD liver may be because sampling took place 4 hours after warm ischemic injury, and that during ANOR the organ was able to initiate repair mechanisms. This would be concordant with the observed dynamic nature of survival mechanisms observed in another ANOR study, albeit focusing on the kidney [22]. Further investigation of the behavior of liver tissue during ANOR is required to investigate this aspect.”

  1. It seems that there is not much difference in band concentration between ANOR-DCD and DBD in HO1, SOD1, SOD2, and HIF1 in Fig. 3 and Fig. 5. Lower SF protein levels in ANOR-DCD than DBD may emphasize the increased protein expression.

We agree with the reviewer that each protein signal, while different, does appear close between DCD and DBD. However indeed there was less total protein as showed by the use of StainFree technology. While confusing, this also highlight the value of SF over classical housekeeping protein normalization: SF evaluates protein loading over the whole gel rather than basing it on a single protein which often is affected by the protocol. We however considered this aspect in the discussion in order to properly consider all aspects of the issue.

It would be better to elaborate on the interpretation of these results to support the conclusion.

 Thank you for these suggestions, we hope our changes meet your expectations in terms of improvement.

2) In Supplementary Figure 2, there is considerable variability in histological evaluation using HE. Though it is mentioned in limitation of discussion part, the fact that the effects of reperfusion injury have not been evaluated is also considered to be a reason why this conclusion is difficult to accept. However, it is impressive that BCLXL, GLUT1 and NOTCH4 at the mRNA level during cold preservation show converse results between DBD and ANOR-DCD. Please explain whether the difference can be associated with the superiority of ANOR-DCD.

 Thank you for this comment, we amended the discussion as follows: “This, combined with the apoptosis data, strongly suggests that ANOR preconditions the organ to better manage reperfusion injury with an enhanced ability to manage cell death as well as to restart energy metabolism and nutrient influx through angiogenesis.”

3) Does the authors think that ANOR after DCD up-regulates stress proteins such as HO1 and HIF1 while DBD is less likely to upregulate them. If so, it would be better to examine the time-dependent change of the stress protein before donation of 4 hours. Is 4 hours normothermic regional perfusion optimal in the examination so far?

 We agree that our results encourage further investigation of ANOR effects on the liver, particularly an in depth investigation of ANOT timing. We amended our conclusion to highlight this, thank you for your suggestion.

4Does "H0" in Fig3 mean "0h"?

Thank you for highlighting this discrepancy, we amended the figure to harmonize it.

Round 2

Reviewer 2 Report

I would like to thank you for the oppotunity to review the manuscript. The authors have taken great care to address all the coments.